# System-2 Mathematical Reasoning via Enriched Instruction Tuning

**Huanqia Cai**                                                    *caihuanqia@gmail.com*
*Tencent*

**Yijun Yang**                                                    *yijun.steven.yang@gmail.com*
*Tencent*
*University of Technology Sydney*

**Zhifeng Li** *[*][†]                                           *zhifeng0.li@gmail.com*
*XIntelligence Technology Co., Ltd.*

**Reviewed on OpenReview:** `https://openreview.net/forum?id=Cl9Uox031k`

## Abstract

Solving complex mathematical problems via system-2 reasoning is a natural human skill, yet it remains a significant challenge for current large language models (LLMs). We identify the scarcity of deliberate multi-step reasoning data as a primary limiting factor. To this end, we introduce **E**nriched **I**nstruction **T**uning (**EIT**), a method that enriches existing human-annotated mathematical datasets by augmenting human-annotated data with AI-generated feedback to create fine-grained reasoning trajectories. These datasets are then used to fine-tune open-source LLMs, enhancing their mathematical reasoning abilities without reliance on any symbolic verification program. Concretely, EIT is composed of two critical steps: Enriching with Reasoning Plan (ERP) and Enriching with Reasoning Step (ERS). The former generates a high-level plan that breaks down complex instructions into a sequence of simpler objectives, while ERS fills in reasoning contexts often overlooked by human annotators, creating a smoother reasoning trajectory for LLM fine-tuning. Unlike existing CoT prompting methods that generate reasoning chains only depending on LLM's internal knowledge, our method leverages human-annotated initial answers as "meta-knowledge" to help LLMs generate more detailed and precise reasoning processes, leading to a more trustworthy LLM expert for complex mathematical problems. In experiments, EIT achieves an accuracy of 84.1% on GSM8K and 32.5% on MATH, surpassing state-of-the-art fine-tuning and prompting methods, and even matching the performance of tool-augmented methods.

## 1 Introduction

Large language models (LLMs) are increasingly recognized as a promising stride towards realizing Artificial General Intelligence (AGI) due to (1) their potential for a wide range of intelligent activities and (2) their scaling up of performance as increased dataset size, model size, and computing budget (Bubeck et al., 2023; Kaplan et al., 2020). Surprisingly, all of the current advances in LLMs are established on an embarrassingly "*simple*" training paradigm, i.e., Transformer decoder + next-token prediction (Radford et al., 2018). Though promising, the most powerful LLMs, e.g., GPT-4 (Achiam et al., 2023), still frequently make ridiculous mistakes when solving fundamental mathematical problems, e.g., multiplication of four digits (Hagendorff et al., 2022), hindering their usage in scenarios requiring mathematical reasoning, such as autonomous driving (Fu et al., 2024), embodied intelligence (Brohan et al., 2023; Yang et al., 2024; Zhou et al., 2024), and teaching assistance (Hicke et al., 2023). The root of this issue is that

---

[*]The work was done during working at Tencent.
[†]Corresponding author.

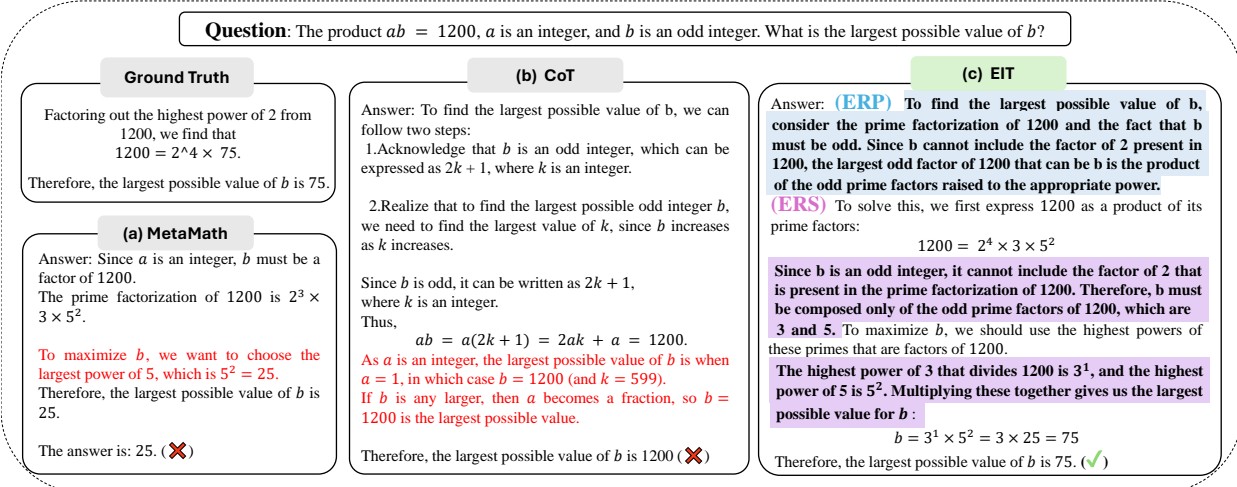

Figure 1: Comparison of LLM's response generated by four different methods on a randomly selected problem from the MATH dataset (Hendrycks et al., 2021). (a) **MetaMath** considers factorization, but the problem-solving plan is wrong and ignores other odd factors. (b) **Chain-of-thought prompting (CoT)** generates a valid plan, but hallucinations occur during the solution process, leading to a trivial calculation error: if $a$ equals 1 and $k$ is 599.5, then $b$ is 1200, which is not an odd integer. (c) The LLM fine-tuned by **our Enriched Instruction Tuning (EIT)** first generates a high-level plan (blue parts) decomposing the original question into a sequence of lower-level objectives and then produces a fine-grained reasoning trajectory (purple parts) **with the guidance from human-provided initial answers**.

LLM-generated responses entirely rely on next-token prediction, a process akin to our intuitive, quick-thinking system-1 reasoning (Daniel, 2017). This is fine for easy tasks but falls short for complex mathematical problems, in which humans usually switch to thoughtful, logical system-2 reasoning, *a slower and more deliberate way of thinking while making our answers more accurate and less biased* (Daniel, 2017)[1].

The deficiency of system-2 mathematical reasoning in LLMs has motivated many recent works that can be divided into two categories: fine-tuning-based and prompting-based methods. Fine-tuning-based methods, such as Phi-GSM (Liu et al., 2023), MetaMath (Yu et al., 2023), MathGenie (Lu et al., 2024), and Orca-Math (Mitra et al., 2024), update LLMs' parameters by distilling privileged LLMs or learning from human's reasoning trajectories. Many of them adopt very complicated augmentation strategies in order to boost performance (see Table 1). For example, Phi-GSM and Orca-Math generate more reliable training data via the ensemble of multiple LLMs. MetaMath and MathGenie adopt several elaborate augmentation strategies, including question rephrasing, backward reasoning, and self-verification, to augment mathematical reasoning data for fine-tuning. Additionally, some methods use enormous amounts of data (e.g., 12M for Phi-GSM) or external tools, e.g., code verification (Wang et al., 2023a) and tree search (Xie et al., 2024; Chen et al., 2024; Brandfonbrener et al., 2024), to correct calculation and reasoning errors. However, such a mixture of many augmentation strategies slows down the run time of these methods and makes causal attributions of performance gains difficult. It even impairs the most attractive property of LLMs (i.e., "scaling law"), as demonstrated in the MetaMath (Yu et al., 2023) and our experiments: *more augmented data may hurt performance*.

Another line of research studies prompting-based methods, such as CoT (Wei et al., 2022), ToT (Yao et al., 2024), ReAct (Yao et al., 2022), and Reflexion (Shinn et al., 2024). They try to induce the potential reasoning capacities of LLMs with only in-context learning. While these methods are conceptually simple and training-free, they often suffer from factual hallucination and error propagation over a long-horizon reasoning trajectory, as demonstrated by Fig. 1 (b). Therefore, in this paper, we ask: *Can LLMs learn to execute system-2 mathematical reasoning on top of the established training paradigm (i.e., next-token prediction)?*

A plausible solution to this problem is learning from human feedback (LHF), which collects a very large-scale dataset of high-quality and multi-step reasoning trajectories from human feedback and then fine-tunes a pre-trained LLM on it via supervised or reinforcement learning (Ouyang et al., 2022; Joshi et al., 2023). Unfortunately, collecting such a dataset is prohibitively expensive and impractical due to (1) the lack of well-trained professional human annotators for

---

[1]The general concept of System-2 reasoning is also employed by other work (LeCun, 2022; Weston & Sukhbaatar, 2023; Yu et al., 2024), which involves generating deliberate and structured thought processes that enable a model (or human) to reason and plan effectively, thereby completing the given task successfully.

complex mathematical problems (Wang et al., 2020) and (2) the biased assessment of annotation's correctness (Huang et al., 2023). Instead of learning from expensive human feedback, an alternative solution is to learn from AI feedback (LAIF) (Lee et al., 2023), which leverages powerful off-the-shelf LLMs, e.g., ChatGPT (Achiam et al., 2023), to automatically produce complete reasoning trajectories and evaluate their quality. Despite promising and scalable, LAIF reintroduces a chicken-and-egg dilemma similar to what this method is trying to address from the beginning: LLMs are not good at system-2 mathematical reasoning, but they are expected to learn such capabilities from their own (potentially low-quality) reasoning trajectories.

To better answer the above question, we combine the strengths of LHF and LAIF in a novel way that avoids their limitations. Concretely, we leverage the synergy of human and AI feedback to produce enriched instruction datasets consisting of fine-grained reasoning trajectories. These datasets are then used to fine-tune open-source LLMs. Such human-AI collaboration achieves a win-win result: high-quality human reasoning trajectories mitigate LLMs' hallucination issue while the almost infinite generative power of LLMs reduces the workload of human annotators. As illustrated in Fig 2, given an instruction-response pair from any existing human-annotated mathematical dataset, our method EIT (standing for **E**nriched **I**nstruction **T**uning) first prompts an LLM to generate a plan decomposing the complex instruction into a sequence of lower-level objectives. With the guidance of the high-level plan, the same LLM is then used to complement reasoning contexts between or within steps omitted in the original response through ERS prompting, leading to a smoother reasoning trajectory for LLM fine-tuning.

We evaluate EIT and compare it with finetuning-based, prompting-based, and tool-augmented methods on two widely used mathematical benchmarks, MATH and GSM8K. Specifically, EIT achieves an accuracy of 32.5% on the MATH dataset and 84.1% on GSM8K, marking an improvement of 2.7% on MATH over MetaMath (Yu et al., 2023). Notably, EIT matches the performance of methods using external tools on the GSM8K benchmark, even outperforming MathCoder (Wang et al., 2023a). We also demonstrated that more fine-grained reasoning trajectories lead to better testing performance, shedding a novel insight that the completeness and quantity of data are equally important.

## 2 Related Work

Mathematical reasoning is a key aspect of human intelligence that enables us to comprehend and make decisions based on numerical data and language (Lu et al., 2022). This cognitive faculty is also an important factor in evaluating the capabilities of LLMs. Mathematical reasoning is still a great challenge for LLMs, which struggle with complex computations and symbolic manipulations. Prompt-based methods are proposed to improve reasoning capabilities. Chain-of-thought prompting (CoT) (Wei et al., 2022; Yao et al., 2024; Shinn et al., 2024; Yao et al., 2022) proposes that LLMs can improve reasoning capabilities by generating reasoning chains by leveraging intermediate natural language reasoning trajectories as prompts. Some recent studies also proposed to select in-context-learning examples, since the chosen examples in prompts have a large impact on the accuracy and stability of reasoning (Rubin et al., 2021; Zhang et al., 2022). (Jin et al., 2024) is a prompting-based method, which claims that more reasoning steps of CoT can achieve better performance. However, only using few-shot examples to prompt LLMs to generate a longer reasoning trajectory may cause severe hallucination issues (Hagendorff et al., 2022).

Fine-tuning is another way to improve reasoning capabilities, which collects a very large-scale dataset of high-quality and multi-step reasoning trajectories from human feedback and then fine-tunes a pre-trained LLM on it via supervised or reinforcement learning (Ouyang et al., 2022; Joshi et al., 2023). Unfortunately, collecting such a dataset is prohibitively expensive and impractical due to (1) the lack of well-trained professional human annotators for complex mathematical problems (Wang et al., 2020) and (2) the biased assessment of annotation's correctness (Huang et al., 2023). Instead of learning from expensive human feedback, an alternative solution is to learn from AI feedback (LAIF) (Lee et al., 2023), which leverages powerful off-the-shelf LLMs, e.g., ChatGPT (Achiam et al., 2023), to automatically produce complete reasoning trajectories and evaluate their quality (Xu et al., 2023; Yu et al., 2023; Liu et al., 2023; Lu et al., 2024; Mitra et al., 2024). Despite being promising and scalable, LAIF cannot directly learn such capabilities from their own reasoning trajectories since they are not good at system-2 mathematical reasoning.

Different from prior works that focus on either LHF or LAIF, we leverage the synergy of human and AI feedback to produce enriched instruction datasets consisting of fine-grained reasoning trajectories. These datasets are then used to fine-tune open-source LLMs, thereby enhancing their own ability to execute system-2 mathematical reasoning without any usage of external tools.

# 3 Enriched Instruction Tuning

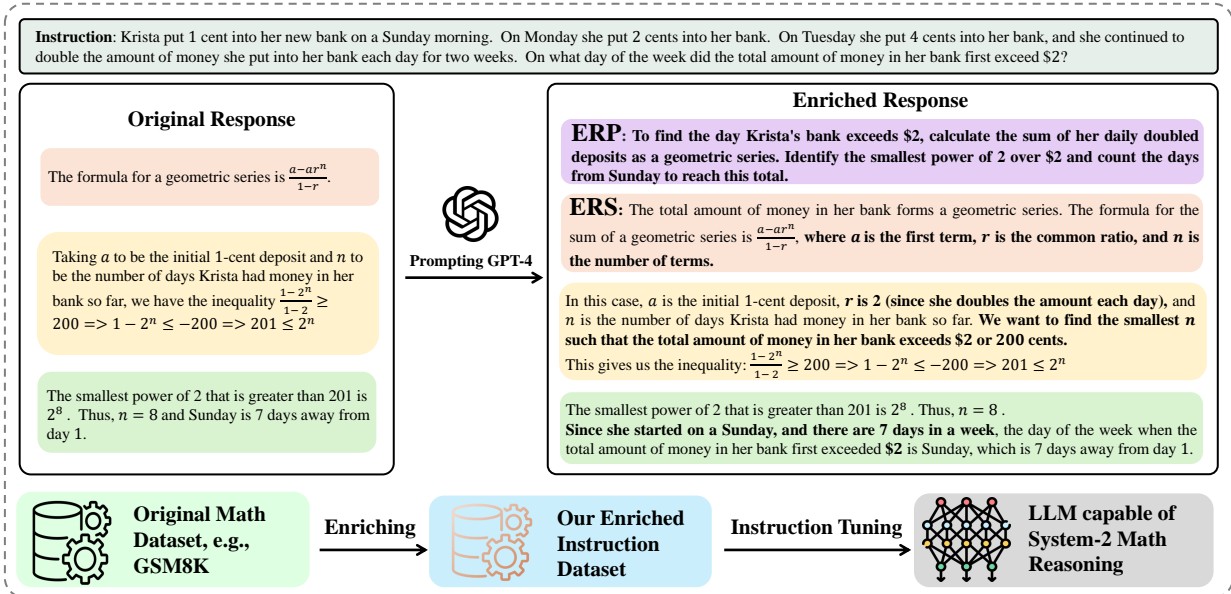

Figure 2: **Pipeline of Enriched Instruction Tuning (EIT)**, which first leverages a privileged LLM, e.g., GPT-4, to produce enriched reasoning steps for the existing mathematical instruction dataset through our proposed ERP and ERS prompting methods, and then trains an LLM on this enriched dataset via instruction tuning. Note that the Original Response provided by human annotators overlooks the **bold** context in Enriched Response, which however is critical to problem-solving.

According to Daniel Kahneman's theory of fast and slow thinking (Daniel, 2017), system-2 reasoning is crucial for humans or machines to solve complex mathematical problems (refer to our detailed discussion in Sec. 1). Based on the existing proven training paradigm, a straightforward method to incorporate such reasoning capabilities into LLMs is to collect a very large-scale dataset consisting of high-quality multi-step reasoning data from human feedback and to fine-tune a pretrained LLM on this dataset via supervised or reinforcement learning (Ouyang et al., 2022; Joshi et al., 2023). However, collecting high-quality multi-step reasoning data is highly challenging: it requires well-trained professional human annotators (Wang et al., 2024), and on the other hand, the reliable and scalable assessment of feedback correctness remains an open problem (Huang et al., 2023). An alternative method is learning from AI feedback (LAIF) (Lee et al., 2023; Yu et al., 2023; Lu et al., 2024), which leverages powerful off-the-shelf LLMs to generate complete reasoning trajectories and evaluate their correctness in lieu of human annotators. Despite promising, LAIF cannot master system-2 reasoning capabilities before generating such data. Hence, how to generate high-quality, detailed reasoning data is still an open challenge.

To this end, we combine the strengths of the two above methods in a novel way that avoids their limitations. Specifically, for existing human-annotated mathematical datasets, we prompt a powerful LLM (e.g., GPT-4) to fill in missing contextual information on high-quality but usually **sparse** human-annotated reasoning trajectories, forming a series of enriched and fine-grained "thought chains", which is then used to fine-tune an open-source LLM. Our approach, named **E**nriched **I**nstruction **T**uning (EIT), comprises two main steps: (1) generation of enriched instruction dataset and (2) LLM fine-tuning, which we will introduce in the following sections respectively. Fig. 2 illustrates the main idea of EIT.

## 3.1 Generation of Enriched Instruction Dataset

Given human-annotated mathematical instruction datasets such as GSM8K (Cobbe et al., 2021) and MATH (Hendrycks et al., 2021), the responses provided by human annotators are typically accurate yet sparse, omitting many of the implicit reasoning steps that humans adopt when solving complex tasks. These omitted steps may include the usage of common sense, identification of variable meanings, determination of causal relationships, and even the pre- and re-planning that humans and many animals implicitly execute in their brains (Daniel, 2017; LeCun, 2022; Hagendorff et al., 2022). As demonstrated in Fig. 1 (b), LLMs fine-tuned on such datasets can only learn to imitate intuitive and

Table 1: Comparison of augmentation strategies: EIT vs. other SOTA methods.

| Methods | Accuracy on GSM8K (%) | Tool | Verifier/Corrector | PPO/DPO | Mixture of Data Aug. Strategies |
|---|---|---|---|---|---|
| Qwen-Chat (Xu et al., 2024) | 76.4 | ✓ | - | ✓ | - |
| MAmmoTH (Yue et al., 2023) | 76.9 | ✓ | ✗ | ✗ | ✓ |
| DeepSeek-Chat (Bi et al., 2024) | 86.7 | ✓ | ✗ | ✓ | ✗ |
| LLEMMA (Azerbayev et al., 2023a) | 62.6 | ✓ | ✗ | ✗ | ✓ |
| ToRA (Gou et al., 2023) | 84.3 | ✓ | ✓ | ✗ | ✗ |
| MathCoder (Wang et al., 2023a) | 83.9 | ✓ | ✗ | ✗ | ✓ |
| WizardMath (Luo et al., 2023) | 81.6 | ✗ | ✗ | ✓ | ✓ |
| MetaMath (Yu et al., 2023) | 82.3 | ✗ | ✗ | ✗ | ✓ |
| CoSC-code (Gao et al., 2024) | 82.3 | ✓ | ✗ | ✗ | ✗ |
| EIT (Ours) | 84.1 | ✗ | ✗ | ✗ | ✗ |

quick-thinking system-1 reasoning. Moreover, relying on human annotators to write out the comprehensive reasoning trajectories for tackling each complex problem is prohibitively expensive and impractical (Wang et al., 2024).

To address this, we leverage powerful off-the-shelf LLMs to enrich the existing mathematical datasets with those missing reasoning contexts through few-shot prompting. The prompts we used for enriching MATH and GSM8K can be found in Example 3.2 and Example 3.3. Example 3.1 provides a detailed comparison between the original responses provided by humans and those enriched using our method. It is worth noting that our proposed method strictly follows the original response without involving additional mathematical or symbolic calculations, maintaining the accuracy of human annotations while significantly mitigating LLMs' hallucinations on long-horizon reasoning trajectories.

Concretely, we design two general prompting methods: (1) Enriching with Reasoning Plan (ERP) and (2) Enriching with Reasoning Step (ERS), which respectively consider the essential roles of planning and reasoning in human's System-2 thinking. Given a human-annotated instruction-response pair from any existing mathematical dataset, ERP prompting leverages an LLM to generate a plan decomposing the complex instruction into a sequence of lower-level objectives (see the red parts in Example 3.1). With the guidance of the semantic plan generated by ERP, the same LLM is then used to fill in missing contextual information and complement reasoning steps omitted in the original response (see the blue parts in Example 3.1) through ERS prompting, which reduces implicitly omitted steps in the response caused by human annotators and results in a more coherent reasoning trajectory for LLM finetuning. Table 2 illustrates the composition of our enriched instruction dataset (i.e., **EITMath**), which is constructed by prompting GPT-4 with our proposed ERP and ERS prompting methods based on MATH (Hendrycks et al., 2021) and GSM8K (Cobbe et al., 2021) datasets.

We also conduct a quality analysis of our EITMath by employing perplexity as a metric to evaluate and calculating the accuracy of final answers(Azerbayev et al., 2023b; Brown et al., 2020). As discussed in Sec. 6.3, our analysis indicates that perplexity is inversely proportional to accuracy. The perplexity of our generated data is significantly lower than that of the original answers, demonstrating the high quality of our generated data. More importantly, we verify output fidelity by comparing final answers (e.g., $\boxed{60}$ in Example 3.1): enriched responses achieve 99.7% alignment with original answers, whereas GPT-4's direct outputs without EIT prompts show only 71.5% accuracy. These findings indicate that our EIT approach substantially enhances the quality of the generated responses and mitigates issues related to hallucinations which lead to wrong final answers.

## 3.2 Finetuning LLMs on Enriched Instruction Dataset

We train an open-source LLM $M$ on our collected EITMath dataset $\mathcal{D}_E = \{(x_t^i, y_r^i)\}_{i=1}^N$ with token-level supervised finetuning (Ouyang et al., 2022). During training, $M$ is expected to produce the corresponding response $y_r$ given the specific instruction $x_t$ as input. The training objective involves next-token prediction, consistent with all decoder-only language models (Radford et al., 2018; Touvron et al., 2023a;b). The likelihood of a target response is modeled as below.

$$P_M\left(y_r^i|x_t^i\right) = \prod_{l=1}^{L} P_M\left(t_l|y_{r,<l}^i, x_t^i\right),  \tag{1}$$

in which $y_{r,<l}^i$ denotes the target response tokens before the current predicted token $t_l$. Eq. 1 can be optimized using the maximum likelihood estimation with mini-batch stochastic gradient descent (Kingma & Ba, 2014).

---

**Example 3.1: Comparison of Original and Enriched Responses**

**Instruction:** A yogurt shop sells four flavors of yogurt and has six different toppings. How many combinations of one flavor and two different toppings are available?

**Original Response:** There are 4 choices of flavor, and $\binom{6}{2} = 15$ ways to choose two of the six toppings. The total number of combinations is $4 \cdot 15 = \boxed{60}$.

**Enriched Response:** To solve this, consider the number of ways to choose one flavor from the four available and then calculate the number of ways to choose two different toppings from the six available. Multiply these two numbers to find the total number of unique combinations.

There are 4 choices of flavor. Since we are choosing 2 toppings out of 6, we can use the combination formula $\binom{n}{r} = \frac{n!}{r!(n-r)!}$, where $n$ is the total number of items, and $r$ is the number of items to choose. In this case, $n = 6$ and $r = 2$, so we have $\binom{6}{2} = 15$ ways to choose two of the six toppings. Finally, since the choice of flavor and the choice of toppings are independent, we multiply the number of choices for each to get the total number of combinations. Therefore, the total number of combinations is $4 \cdot 15 = \boxed{60}$.

---

**Example 3.2: Prompt for Enriching MATH (ERP and ERS)**

You are an answer enricher. I hope you will refine your answers based on the questions and answers provided and avoid jumping your thinking. Pay attention to filling the missing contexts and logical gaps in existing answers. Besides, be careful not to change the original pathways or modify formulas and results, as well as mathematical symbol formats. The first step is to give the idea of the answer, which helps the respondent to solve the question better. Finally, solve the problem.

**question**: Krista put 1 cent into her new bank on a Sunday morning. On Monday she put 2 cents into her bank. On Tuesday she put 4 cents into her bank, and she continued to double the amount of money she put into her bank each day for two weeks. On what day of the week did the total amount of money in her bank first exceed $2?

**answer**: The formula for a geometric series is $\frac{a-ar^n}{1-r}$. Taking $a$ to be the initial 1-cent deposit and $n$ to be the number of days Krista had money in her bank so far, we have the inequality

$$\frac{1-2^n}{1-2} \geq 200 \Rightarrow 1 - 2^n \leq -200 \Rightarrow 201 \leq 2^n.$$

The smallest power of 2 that is greater than 201 is $2^8$. Thus, $n = 8$ and $\boxed{\text{Sunday}}$ is 7 days away from day 1.

**enriched answer**: <planning> To find the day Krista's bank exceeds $2, calculate the sum of her daily doubled deposits as a geometric series. Identify the smallest power of 2 over $2 and count the days from Sunday to reach this total.

<solution> The total amount of money in her bank forms a geometric series. The formula for the sum of a geometric series is $\frac{a-ar^n}{1-r}$, where $a$ is the first term, $r$ is the common ratio, and $n$ is the number of terms. In this case, $a$ is the initial 1-cent deposit, $r$ is 2 (since she doubles the amount each day), and $n$ is the number of days Krista had money in her bank so far. We want to find the smallest $n$ such that the total amount of money in her bank exceeds $2, or 200 cents. This gives us the inequality:

$$\frac{1-2^n}{1-2} \geq 200 \Rightarrow 1 - 2^n \leq -200 \Rightarrow 201 \leq 2^n.$$

The smallest power of 2 that is greater than 201 is $2^8$. Thus, $n = 8$. Since she started on a Sunday, and there are 7 days in a week, the day of the week when the total amount of money in her bank first exceeded $2 is $\boxed{\text{Sunday}}$, which is 7 days away from day 1.

**question**: {Q}
**answer**: {A}
**enriched answer**:<planning>

---

# 4 Experiments

In order to verify the effectiveness of EIT, we collect EITMath (see Table 2) and use it as the training dataset to fine-tune open-source LLaMA-2 (Touvron et al., 2023b) 13B and 70B models and evaluate their testing performance on two popular benchmarks, MATH (Hendrycks et al., 2021) and GSM8K (Cobbe et al., 2021).

---

**Example 3.3: Prompt for Enriching GSM8K (ERP and ERS)**

You are an answer enricher. I hope you will refine your answers based on the questions and answers provided, avoid jumping your thinking. Pay attention to filling the missing contexts and logical gaps in existing answers. Besides, be careful not to change the original pathways or modify formulas and results, as well as special symbol expressions, such as 4 x 15 = «4*15=60»60. The first step is to give the idea of the answer, which helps the respondent to solve the question better. Finally, solve the problem.
**question**:Albert is wondering how much pizza he can eat in one day. He buys 2 large pizzas and 2 small pizzas. A large pizza has 16 slices and a small pizza has 8 slices. If he eats it all, how many pieces does he eat that day?
**answer**:He eats 32 from the largest pizzas because **2 x 16 = «2\*16=32»32**
He eats 16 from the small pizza because **2 x 8 = «2\*8=16»16**
He eats 48 pieces because **32 + 16 = «32+16=48»48**
#### 48
**enriched answer**: <planning>To solve this, calculate the total number of slices in the large pizzas and the small pizzas separately, then add those numbers together to find the total number of slices Albert will eat.
<solution>Albert purchases two large pizzas and two small pizzas for his day of pizza eating. Each large pizza is divided into 16 slices, thus he has **2 x 16 = «2\*16=32»32** slices from the large pizzas.
Similarly, each small pizza is divided into 8 slices. Therefore, from the small pizzas, he has **2 x 8 = «2\*8=16»16** slices.
By adding up the slices from both the large and small pizzas, we find that Albert eats a total of **32 + 16 = «32+16=48»48** slices of pizza in one day. So, if Albert consumes all the pizzas he bought, he will have eaten 48 slices of pizza in a single day.
#### 48
**question**: {Q}
**answer**: {A}
**enriched answer**:<planning>

---

Table 2: Our EITMath Dataset.

| Datasets | w/ ERP & ERS |
|---|---|
| Enriched MATH (Hendrycks et al., 2021) | 7.5k |
| Enriched GSM8K (Cobbe et al., 2021) | 7.5k |

Table 3: Ablation study of ERP, ERS, and MetaMath on MATH and GSM8K datasets.

| ERP | ERS | MetaMath | MATH | GSM8K |
|---|---|---|---|---|
| | | | 14.9 | 67.3 |
| ✓ | | | 17.2 | 72.0 |
| | ✓ | | 19.5 | 76.3 |
| | | ✓ | 19.0 | 74.2 |
| ✓ | ✓ | | 21.3 | 78.8 |
| ✓ | ✓ | ✓ | **22.6** | **81.1** |

## 4.1 Datasets

**MATH** dataset collects a total of 12,500 competition-level mathematics problems, which are partitioned into 7,500 for training and 5,000 for testing. Each of them is accompanied by a step-by-step solution and concludes with a distinct final answer, which is formatted for a straightforward comparison with model-generated solutions. Moreover, the MATH dataset spans a broad spectrum of subjects and difficulty levels, including seven categories: Prealgebra, Algebra, Number Theory, Counting and Probability, Geometry, Intermediate Algebra, and Precalculus.

**GSM8K** dataset is a diverse collection of grade school mathematical word problems, as recognized for its high quality. While it is generally considered less challenging than the MATH dataset, it provides more fine-grained step-level solutions with basic arithmetic operations (addition, subtraction, multiplication, division). Following the same setting as prior works, we use 7,473 and 1,319 problems for training and testing, respectively.

**EITMath** Table 2 illustrates the composition of our collected EITMath dataset, which contains 15k problems (7.5k from MATH and 7.5k from GSM8K) and the corresponding answers enriched with our proposed ERP and ERS prompting methods. The detailed construction pipeline can be found in Sec. 3.1 and illustrated in Fig. 2.

## 4.2 Experimental Setup

We use open-source LLMs LLaMA-2 as the base model for fine-tuning. GPT-4-1106-preview is used to generate enriched responses for constructing EITMath. Following the prior work (Yu et al., 2023; Xu et al., 2023), we adopt the

AdamW optimizer to train the model with 3 epochs, and the learning rate is set to 2e-5. The batch size is 32 for the 70B model and 128 for the 13B model. 32 A100 GPUs are used to fine-tune the above models.

We conduct the ablation study on MATH and GSM8K datasets based on LLaMA-2-70B in Table 3. To make a fair comparison, we use the same dataset size of 7.5k as MATH and GSM8K's training dataset. Furthermore, we combine EIT with MetaMath (Yu et al., 2023), a method focusing on augmenting questions themselves, to explore whether combining them results in better performance. Considering other SOTA methods in Table 4 using larger datasets (e.g., MetaMath (Yu et al., 2023) for 395k, MAmmoTH (Yue et al., 2023) for 260k), we also expand EITMath to 70k in order to make a fair comparison. To this end, we construct such a dataset with more diverse responses by sampling GPT-4's output distribution with different temperature coefficients, which slightly improves EIT's performance.

## 5 Ablation Study

To evaluate the effectiveness of our proposed Enriching with Reasoning Planning (ERP) and Enriching with Reasoning Step (ERS) prompting methods, we conducted the ablation study on MATH and GSM8K datasets based on LLaMA-2-70B. We construct ERP and ERS datasets separately by prompting GPT4. The prompts that we used for ablation study can be found in Appendix A.4.1 and Appendix A.4.2

**Effects of ERP** Table 3 shows the performance of baseline methods and the enhancements achieved by finetuning LLMs on enriched datasets with ERP prompting. The baseline exhibits an accuracy of 14.9% on the MATH dataset and 67.3% on the GSM8K dataset. Incorporating ERP results in a notable accuracy increase to 17.2% on MATH and 72.0% on GSM8K, corresponding to improvements of 2.3% and 4.7%, respectively. These results underscore the significant role of ERP in improving LLM's reasoning capabilities. By generating a high-level plan, ERP effectively decomposes complex instructions into lower-level objectives, thereby enhancing the coherence and direction of the ensuing reasoning process.

**Effects of ERS** The performance of the ERS is also quantified in Table 3. ERS leads to a rise in accuracy from 14.9% to 19.5% on the MATH dataset, marking a 4.6% improvement. Similarly, on the GSM8K dataset, the LLM's accuracy is propelled from 67.3% to 76.3%, obtaining a 9% improvement. This significant improvement can be attributed to the ERS's ability to bridge the gaps in implicit contextual information or steps omitted by human annotators, thereby improving LLM's reasoning capabilities. Notably, the improvement of GSM8K is greater than that of MATH, with an increase of 9%, which shows that ERS can more significantly improve performance on easier mathematical problems, such as those from the GSM8K, compared to the more challenging MATH.

Finally, when incorporated with ERP and ERS, our EIT yields the most pronounced performance gains, culminating in a peak accuracy of 21.3% on the MATH dataset and 78.8% on the GSM8K dataset.

**Effects of Combining EIT and MetaMath** There are various methods that emphasize the augmentation of questions with multiple strategies (e.g., MetaMath, WizardMath), while our EIT focuses on enriching the responses. Given the complementary nature of these approaches, we explored their combined effect by conducting ablation experiments with MetaMath and incorporating our proposed EIT. To ensure a fair comparison, we curated a subset of the augmented MATH data from MetaMathQA, matching the size of the original MATH dataset. A similar approach was taken with the GSM8K dataset. The results, as detailed in Table 3, indicate a notable performance enhancement when both question augmentation and response enrichment strategies are employed. Specifically, the model exhibits an incremental gain of 1.3% on the MATH dataset and 2.3% on GSM8K when compared to the response enrichment-only configuration. Moreover, the improvements are even more pronounced when EIT is applied to question augmentation-only configuration, with increases of 3.6% on MATH and 6.9% on GSM8K. These findings confirm that augmenting both questions and responses can substantially elevate the model's reasoning capabilities, with the combined approach yielding superior accuracy. Furthermore, the data suggests that our EIT contributes more effectively to performance enhancement than question augmentation when the volume of data is held constant.

Table 4: Comparison of testing accuracy to existing LLMs on GSM8K and MATH testing sets. † means that external tools are used. ‡ means that the MetaMath is fine-tuned by QLoRA with the batch of 128 from (Yu et al., 2023), while the full fine-tuned version is from (Wang et al., 2023b).

| Methods | Model Size | MATH | GSM8K |
|---|---|---|---|
| LLaMA-1 (Touvron et al., 2023a) | 13B | 3.9 | 17.8 |
| LLaMA-2 (Touvron et al., 2023a) | 13B | 3.9 | 28.7 |
| MPT (Team et al., 2023) | 30B | 3.1 | 15.2 |
| Falcon (Penedo et al., 2023) | 40B | 2.5 | 19.6 |
| Vicuna (Chiang et al., 2023) | 13B | - | 27.6 |
| WizardMath (Luo et al., 2023) | 13B | 14.0 | 63.9 |
| MetaMath (Yu et al., 2023) | 13B | 22.4 | 72.3 |
| EIT | 13B | **23.0** | **73.1** |
| LLaMA-1 (Touvron et al., 2023a) | 65B | 10.6 | 50.9 |
| LLaMA-2 (Touvron et al., 2023a) | 70B | 13.5 | 56.8 |
| Platypus-2 | 70B | 15 | 45.9 |
| WizardMath (Luo et al., 2023) | 70B | 22.7 | 81.6 |
| MetaMath‡ (Yu et al., 2023) | 70B | 26.6 | 82.3 |
| MetaMath (Wang et al., 2023b) | 70B | 29.8 | 80.4 |
| Qwen-Chat (Xu et al., 2024) | 72B | 31.8 | 76.4 |
| EIT | 70B | **32.5** | **84.1** |
| PAL(LLaMA-2)† (Gou et al., 2023) | 70B | 18.3 | 55.2 |
| MAmmoTH† (Yue et al., 2023) | 70B | 41.8 | 76.9 |
| MathCoder† (Wang et al., 2023a) | 70B | 45.1 | 83.9 |
| ToRA† (Gou et al., 2023) | 70B | 49.7 | 84.3 |
| DeepSeek-Chat† (Bi et al., 2024) | 67B | 51.1 | 86.7 |
| CoSC-code† (Gao et al., 2024) | 34B | 53.5 | 82.3 |

# 6 Comparison with State of the Art

## 6.1 Results on MATH and GSM8K

As shown in Table 4, compared to the same open-source LLMs that do not use any external tools, such as code verification, EIT achieves the best performance. Our EIT-70B model achieves a leading accuracy of 32.5%, with a 2.7% improvement over MetaMath-70B and more than double the accuracy of the baseline LLaMA-2-70B model. For the 13B model, our EIT model also achieves the best performance among models of the same type, achieving 23% accuracy. These results underscore the superiority of our ERP and ERS methodologies in comparison to existing approaches. To further analyze the improvement of our method, we show the accuracy of subtopics on MATH in Table 5. Our EIT exceeds WizardMath and MetaMath across all subtopics. Notably, EIT achieves 15.4% on the most challenging topic, Precalculus, surpassing MetaMath 4.6% and WizardMath 2.8%.

On the GSM8K benchmark, our EIT-70B records an 84.1% accuracy, with a 2.5% improvement over WizardMath and 1.8% over MetaMath with the same parameters. Furthermore, the 13B model demonstrates an even more remarkable performance, surpassing WizardMath by 9.2% and MetaMath by 0.8%. It is noteworthy that the performance of EIT-70B on GSM8K is comparable to that of models utilizing external computational tools, and even outperforms MathCoder. This underscores the significant potential of harnessing the intrinsic reasoning faculties of LLMs without reliance on additional aids.

Table 5: Comparison of testing accuracy (%) of 70B models on MATH Subtopics.

| Methods | Intermediate Algebra | Precalculus | Geometry | Number Theory | Counting | Prealgebra | Algebra | Overall |
|---|---|---|---|---|---|---|---|---|
| WizardMath | 7.1 | 12.6 | 15.7 | 16.3 | 17.3 | 41.7 | 33.3 | 22.7 |
| MetaMath | 14.28 | 10.8 | 20.9 | 23.7 | 24.3 | 46.9 | 44.56 | 29.8 |
| EIT | **15.0** | **15.4** | **25.9** | **26.7** | **28.7** | **50.0** | **47.1** | **32.5** |

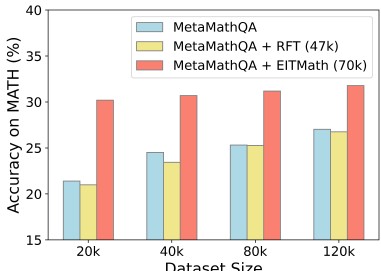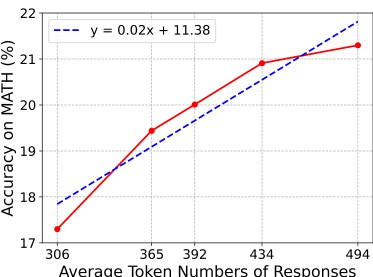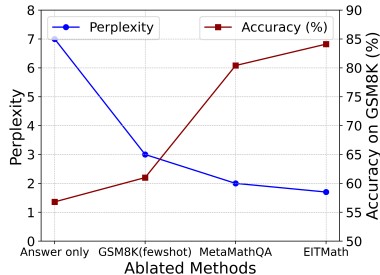

Figure A.1: *left*: Scaling up of performance on MATH as adding our EITMath dataset for LLM fine-tuning. Different colored bars represent MetaMathQA combined with Rejection sampling Fine-Tuning (RFT) and EITMath as the training set. *middle*: Scaling up of performance on MATH as more fine-grained reasoning steps created for fine-tuning. We use average tokens (more is better) of the response to measure its granularity. *right*: Perplexity and accuracy of different methods on GSM8K. Following (Yu et al., 2023), the perplexity is calculated using under-finetuned LLaMA-2-7B, and the accuracy is reported based on our fine-tuned LLaMA-2-70B model on these datasets. **It is clear that our EITMath has less perplexity compared to other mathematical datasets, which leads to better performance.**

## 6.2    Analysis from a Scaling Law Perspective

**Better Performance with More Data**   The scaling law of dataset size is the most attractive property of LLMs. However, recent findings from MetaMath demonstrate that more data is not always better. By combining the existing augmented dataset Rejection sampling Fine-Tuning (RFT) (Yuan et al., 2023) with MetaMathQA of different scales for finetuning, we found that more augmented data hurts performance (Yu et al., 2023). In contrast, our method presents a different perspective. As shown in Fig. A.1 *left*, we conducted a series of finetuning experiments on the LLaMA-2-70B model using different subsets of MetaMathQA data—20k, 40k, 80k, and 120k samples—combined with our EITMath dataset. When compared to the integration with RFT, our approach yields a markedly better performance. The inclusion of RFT consistently led to a decline in performance across all data scales. On the other hand, the incorporation of EITMath not only enhanced the model's performance but also demonstrated the benefits of additional data scale positively with performance, in line with the scaling law. This indicates that the quality of data augmentation and the strategic approach to dataset construction are pivotal factors. Our findings affirm that, based on EIT, more data can indeed lead to better performance in training LLMs.

**Better Performance with More Fine-grained Reasoning Steps**   Observing the performance improvement brought by scaling dataset size, we raise a question: whether more fine-grained reasoning steps can also enhance performance? To explore this, we curated multiple datasets derived from the MATH benchmark, each featuring a different average token number of responses. These datasets were constructed using the same strategy as EITMath. We control the granularity of step-by-step reasonings by adjusting the prompts used when requesting responses from GPT-4. This includes specifying a desired number of tokens for the reply and carefully selecting few-shot examples that demonstrate the desired level of detail. By increasing the granularity of enriched responses, as depicted in Fig. A.1 *middle*, we observed a positive correlation between model performance and tokens for finetuning.

## 6.3    Analysis from a Perplexity Perspective

Prior work (Yu et al., 2023) demonstrates that the simplicity of the data can better elicit the reasoning abilities of LLMs, because such data are easier for the models to learn from, leading to improved performance. The simplicity can be measured by perplexity (Marion et al., 2023). We calculate the average perplexity across all tokens in generated responses from our EITMath using the LLaMA-2-7B model without any additional fine-tuning. The results show that the perplexity of EITMath is lower than all other methods in Fig. A.1 *right*, and the LLM fine-tuned on EITMath also achieves the highest accuracy on GSM8K.

## 6.4    Comparison between EIT and CoT

Our method significantly differs from the Chain-of-Thought (CoT) prompting method in several key aspects. While CoT relies on generating intermediate natural language trajectories to improve reasoning capabilities using the model's "internal knowledge", EIT leverages human-annotated initial answers as "meta-knowledge" to guide the model in

generating more detailed and smoother reasoning, making the task more similar to a fill-in-the-blank question. This reduces the complexity of the task, as the model only needs to elaborate on the given answer rather than generate the entire reasoning chain from scratch. Additionally, compared to the standard CoT prompting method that requires complex and carefully selected few-shot examples, EIT's data construction process simplifies the prompting process by providing clear and structured initial answers, reducing the reasoning load on the model.

To further validate the effectiveness of our EIT, we conducted a comparative analysis on fine-tuning and prompting methods, especially the CoT method. As illustrated in Table 6, the performance of prompting with few-shot examples including ERP and ERS is inferior to that of standard CoT prompting. This is because ERP and ERS place higher demands on the model's capabilities, like requiring a more nuanced understanding of complex instructions and stronger reasoning abilities to generate more detailed and smoother reasoning trajectories. Conversely, fine-tuning with standard CoT performs significantly better than standard CoT prompting, particularly on the GSM8K dataset, where it outperforms CoT prompting by 11.5%. This improvement can be attributed to fine-tuning's ability to stimulate the model's CoT capabilities while avoiding the instability introduced by complex prompts.

Notably, our EIT achieves the best performance, with the accuracy of 32.5% on the MATH dataset and 84.1% on the GSM8K dataset, representing nearly a 25.3% and 51.2% improvement compared to prompting with ERP and ERS and over 15% improvement than fine-tuning with standard COT. Unlike data constructed using the CoT method, EIT further leverages human-annotated initial answers as "meta-knowledge", which helps LLMs generate more detailed and smoother reasoning reasoning trajectories. This approach effectively mitigates the accumulation of model hallucinations as the length of inference increases, thereby ensuring the accuracy of our constructed data.

Table 6: Comparison of accuracy (%) of prompting and fine-tuning methods.

| Methods | Model Size | MATH | GSM8K |
|---|---|---|---|
| LLaMA-2 (prompting with standard CoT) | 70B | 14.4 | 57.8 |
| LLaMA-2 (prompting with few-shot examples including ERP and ERS) | 70B | 7.2 | 32.9 |
| LLaMA-2 (fine-tuning with standard CoT) | 70B | 14.9 | 69.3 |
| EIT (ours) | 70B | **32.5** | **84.1** |

## 6.5 Analysis on Case Study

Example 6.1 shows a randomly selected case including responses generated by SFT (supervised fine-tuning using publicly available training sets), MetaMath, and our EIT on the test set of MATH. It is obvious that the first half of the solution from SFT is correct, but there was an error when calculating $\frac{13}{4} \times 36$ due to merging calculations. As for MetaMath, unlike SFT, it divides $3\frac{1}{4}$ yards into 3 and $\frac{1}{4}$ to calculate inches respectively and find their sum finally. MetaMath aims to convert 1/4 yards into inches by utilizing the yard-foot and foot-inch relationships. However, it neglects the relationship between yards and foot, which leads to hallucinations finally. In contrast to the above methods, firstly, our EIT illustrates ERP accurately, guiding the subsequent steps to convert the mixed number to an improper fraction and then multiply by the conversion factor that relates yards to inches. Secondly, EIT demonstrates ERS following the guidance of ERP, which illustrates the conversion factor as the base knowledge and multiplies the converted improper fraction with the conversion factor. Furthermore, EIT divides $\frac{13}{4} \times 36$ into $13 \times 36$ and then divides by 4, which avoids the calculation error that occurs in SFT. As a result, EIT gets the final correct answer. More cases can be found in Example A.1 and Example A.2.

## 7 Conclusion

In this paper, we aim to build an LLM capable of system-2 mathematical reasoning. To achieve this, we introduce Enriched Instruction Tuning (EIT), which leverages the synergy between human and AI feedback to produce enriched instruction datasets consisting of fine-grained reasoning trajectories. These datasets are then utilized to fine-tune open-source LLMs, thereby boosting their inherent ability to execute system-2 mathematical reasoning without any usage of external tools. Compared with finetuning-based, prompting-based, and tool-augmented methods, LLMs fine-tuned by EIT achieve competitive performance on two widely used mathematical benchmarks, i.e., MATH and GSM8K.

---

**Example 6.1: Case Study in Test Set of MATH**

**Question:** How many inches are in $3\frac{1}{4}$ yards? (Ground truth is 117)

**SFT:** To find out how many inches are in $3\frac{1}{4}$ yards, we can use the conversion factor that 1 yard equals 36 inches.
First, we convert the mixed number $3\frac{1}{4}$ yards to an improper fraction. This gives us $\frac{12}{4} + \frac{1}{4} = \frac{13}{4}$ yards.
Next, we multiply this fraction by the conversion factor, which is 36 inches per yard. This gives us:

$$\frac{13}{4} \cdot 36 = 216 \text{ inches.}$$

So, there are $\boxed{216}$ inches in $3\frac{1}{4}$ yards. (✗)

**MetaMath:** There are 36 inches in 1 yard. So, there are $3 \times 36 = 108$ inches in 3 yards. There are 12 inches in 1 foot.
So, there are $12 \times 1 = 12$ inches in 1 foot. Therefore, there are $12 \times 1 = 12$ inches in $1\frac{1}{4}$ yards.Adding the inches in 3 yards and $1\frac{1}{4}$ yards, we get $108 + 12 = \boxed{120}$ inches. The answer is: 120. (✗)

**EIT:** We need to convert $3\frac{1}{4}$ yards into inches. To solve the problem, convert the mixed number to an improper fraction to represent the yards, then multiply by the conversion factor that relates yards to inches. Simplify the resulting fraction to find the number of inches.
To convert yards to inches, we use the conversion factor that 1 yard equals 36 inches. We start by expressing $3\frac{1}{4}$ yards as a mixed number and then convert it to an improper fraction: $3\frac{1}{4} = \frac{13}{4}$. Now, we multiply the fraction by the conversion factor:

$$\frac{13}{4} \times 36 = \frac{13 \times 36}{4}.$$

We can simplify this by multiplying 13 by 36, which gives us 468, and then dividing by 4: $\frac{468}{4} = 117$. Therefore, $3\frac{1}{4}$ yards are equal to $\boxed{117}$ inches. The answer is: 117. (✓)

---

Moreover, we find that more fine-grained reasoning trajectories consistently lead to higher testing accuracy, suggesting that the granularity and size of reasoning datasets are likely to be equally important for training a math LLM expert.

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

# A    Appendix

---

**Example A.1: Case Study in Test Set of MATH**

**Question:** A right triangle has leg lengths of 8 and 15 units. What is the length, in units, of the median to the hypotenuse?

Express your answer as a common fraction. (Ground truth is $\boxed{\dfrac{17}{2}}$ )

**SFT:** Let one of the legs of the triangle be $AB$, and let the hypotenuse be $AC$. Let the length of the median be $AM$. Since $\triangle ABM$ is a right triangle, we can use the Pythagorean Theorem to find $AM$:

$$AB^2 + BM^2 = AM^2$$
$$64 + BM^2 = AM^2$$
$$BM^2 = AM^2 - 64$$
$$BM = AM\sqrt{1 - \frac{64}{AM^2}}$$
$$BM = \frac{15}{2}\sqrt{1 - \frac{64}{15^2}}$$
$$BM = \frac{15}{2} \cdot \frac{1}{2} = \boxed{\frac{15}{4}}$$

(✗)

**MetaMath:** First, we need to find the length of the hypotenuse. Using the Pythagorean theorem, we have $8^2 + 15^2 = c^2$, where $c$ is the length of the hypotenuse.Simplifying, we get $64 + 225 = c^2$, or $c^2 = 289$. Taking the square root of both sides, we find $c = 17$. Now, we need to find the length of the median to the hypotenuse. The median to the hypotenuse is the line segment that connects the midpoint of the hypotenuse to one of the legs of the triangle. Since the triangle is a right triangle, the median to the hypotenuse is also the altitude of the triangle. Using the Pythagorean theorem again, we have $8^2 + (17/2)^2 = 15^2$. Simplifying, we get $64 + 289/4 = 225$. Multiplying both sides by 4, we get $256 + 289 = 900$. Subtracting 256 from both sides, we find $289 = 644$. Taking the square root of both sides, we get $\sqrt{289} = \sqrt{644}$. Simplifying, we find $17 = 25$. Therefore, the length of the median to the hypotenuse is $\boxed{\dfrac{17}{25}}$. The answer is: $\frac{17}{25}$ (✗)

**EIT:** The solution involves recognizing that the median to the hypotenuse of a right triangle is half the length of the hypotenuse. By using the Pythagorean theorem, we can find the length of the hypotenuse and then divide it by 2 to get the length of the median. First, we use the Pythagorean theorem to find the length of the hypotenuse. The theorem states that in a right triangle, the square of the length of the hypotenuse (the side opposite the right angle) is equal to the sum of the squares of the lengths of the other two sides. For our triangle with leg lengths of 8 and 15 units, the hypotenuse length $h$ is given by:

$$h = \sqrt{8^2 + 15^2} = \sqrt{64 + 225} = \sqrt{289} = 17.$$

The median to the hypotenuse of a right triangle is always half the length of the hypotenuse. Therefore, the length of the median $m$ is:

$$m = \frac{h}{2} = \frac{17}{2} = 8.5.$$

Thus, the length of the median to the hypotenuse, expressed as a common fraction, is $\boxed{\dfrac{17}{2}}$ units. The answer is: $\frac{17}{2}$ (✓)

---

---

**Example A.2: Case Study in Test Set of MATH**

**Question:** At a driver education class, the students were told to put their hands on the steering wheel on "3" and "9", just as on the face of a clock. If, instead, the students placed their hands on the "2" and the "10", by how many degrees would the angle between the hands (with the center of the wheel at the vertex of the angle) decrease? (Ground truth is $\boxed{60°}$)

**SFT:** The angle between the hands is $360° - 90° = 270°$ at "3" and "9". If the students place their hands on the "2" and the "10", the angle between the hands is $360° - 20° = 340°$. Therefore, the angle between the hands decreases by $340° - 270° = \boxed{70}$ degrees. (✗)

**MetaMath:** The angle between the hands on "3" and "9" is 90 degrees.The angle between the hands on "2" and "10" is 180 degrees. So, the angle between the hands on "2" and "10" is $180 - 90 = 90$ degrees less than the angle between the hands on "3" and "9".Therefore, the angle between the hands on "2" and "10" would decrease by $\boxed{90}$ degrees.The answer is: 90 (✗)

**EIT:** The idea is to calculate the initial angle between the 3 and 9 hands and then calculate the new angle between the 2 and 10 hands. The difference between these two angles will give us the decrease in the angle.

Initially, the hands are at the 3 and 9 positions, which are 6 hours apart on the clock. The angle between any two adjacent numbers on a clock is $360°/12 = 30°$. Therefore, the initial angle between the 3 and 9 hands is $6 \times 30° = 180°$.

When the hands are moved to the 2 and 10 positions, they are 8 hours apart. Again, using the fact that the angle between any two adjacent numbers on a clock is $30°$, the angle between the 2 and 10 hands is $8 \times 30° = 240°$.

The decrease in the angle is the difference between the initial angle and the new angle, which is $180° - 240° = -60°$. Since we are looking for a decrease, we take the absolute value, which is $|-60°| = 60°$.

Therefore, the angle between the hands decreases by $\boxed{60°}$.

Note: The negative sign in the calculation indicates that the angle is decreasing, and taking the absolute value gives us the positive decrease in degrees.The answer is: $60°$. (✓)

## A.1 More Results on Self-Consistency

Self-consistency (Wang et al., 2022) helps to boost models' performance, especially on arithmetic benchmarks such as GSM8K. We investigate the impact of scaling decoding samples and observe that EIT outperforms other methods, as demonstrated in Table 7. ToRA, with external tools, performs comparably to EIT without self-consistency decoding. Applying self-consistency decoding with 50 samples, ToRA attains 4% improvements from 84.1% to 88.3%. In contrast, our EIT outperforms ToRA and achieves 88.4% accuracy with only 5 samples. Notably, our EIT reaches 90.4% accuracy with 50 decoding samples, representing a 6.3% improvement and surpassing ToRA by 2.1%. These results underscore that EIT demonstrates superior reasoning capabilities and robustness compared to existing methods.

Table 7: Comparison of testing accuracy (%) of more results with self-consistency. SC (k=50) denotes self-consistency/majority voting decoding with 50 samples. Notably, EIT + SC (k=5) surpasses ToRA + SC (k=50).

| Methods | Model Size | GSM8K |
|---|---|---|
| ToRA Gou et al. (2023) | 70B | 84.3 |
| ToRA + SC (k=50) | 70B | 88.3 |
| EIT | 70B | 84.1 |
| EIT + SC (k=5) | 70B | **88.4** |
| EIT + SC (k=10) | 70B | **89.9** |
| EIT + SC (k=50) | 70B | **90.4** |

## A.2 Discussion on Datasets Only with Final Answers

Considering that a small portion of datasets contains only the final answers without accompanying reasoning, generating enriched instruction datasets becomes more challenging. We plan to extend our method to such datasets in future work. The potential approach involves a two-stage process. First, LLMs will be prompted to generate sparse responses, those that lead to the correct final answer will kept for the second stage. In the second stage, we enrich the reasoning trajectories progressively in several iterations and use the COT-decoding (Wang & Zhou, 2024) to choose reasoning trajectories with higher confidence.

### A.3 Pattern Analysis on The Enriched Responses

We performed a pattern analysis on the token length of responses across increasing difficulty levels on the MATH dataset. The results are shown in Table 8. Our analysis indicates that as the difficulty increases, the length of the responses generated by EIT also increases. Since the responses of simpler questions tend to have more concise explanations and can overlook some steps, the token length for human annotations is relatively short, resulting in larger length ratios for EIT, vice versa. This result demonstrates that EIT can adapt to the difficulty level of the problem and the human annotations, and adaptively enrich necessary details, highlighting the effectiveness of our approach.

Table 8: Pattern analysis on the token length of responses across increasing difficulty levels on the MATH dataset

| Difficulty Level | 1 | 2 | 3 | 4 | 5 |
|---|---|---|---|---|---|
| Human-Annotated | 57 | 54 | 68 | 161 | 152 |
| EIT | 240 | 248 | 273 | 345 | 362 |
| Length Ratio (EIT / Human-Annotated) | 4.2 | 4.6 | 4.0 | 2.1 | 2.4 |

### A.4 Prompt for Ablation Study of ERP and ERS

We present specific instructions and example few-shot prompts of our EIT's ablation study.

#### A.4.1 Prompt for Enriching MATH (ERP)

The prompt of ERP for generating a high-level plan for MATH dataset can be found in Example A.3, which is used for ablation study in Sec.5.

---

**Example A.3: Prompt for Enriching MATH (ERP)**

You are an expert answer assistant who helps respondents understand the given answer. Based on the question and answer I provide, your task is to generate a high-level outline or plan of the answer. This plan will guide the respondent in formulating their own detailed response and serve as a preparatory framework for addressing the question effectively.

**question**: Krista put 1 cent into her new bank on a Sunday morning. On Monday she put 2 cents into her bank. On Tuesday she put 4 cents into her bank, and she continued to double the amount of money she put into her bank each day for two weeks. On what day of the week did the total amount of money in her bank first exceed $2?

**answer**: The formula for a geometric series is $\frac{a-ar^n}{1-r}$. Taking $a$ to be the initial 1-cent deposit and $n$ to be the number of days Krista had money in her bank so far, we have the inequality

$$\frac{1-2^n}{1-2} \geq 200 \Rightarrow 1-2^n \leq -200 \Rightarrow 201 \leq 2^n.$$

The smallest power of 2 that is greater than 201 is $2^8$. Thus, $n = 8$ and $\boxed{\text{Sunday}}$ is 7 days away from day 1.

**planning**:To find the day Krista's bank exceeds $2, calculate the sum of her daily doubled deposits as a geometric series. Identify the smallest power of 2 over $2 and count the days from Sunday to reach this total.

**question**: {Q}

**answer**: {A}

**planning**:

---

#### A.4.2 Prompt for Enriching MATH (ERS)

The prompt of ERS for enriching MATH dataset can be found in Example A.4, which is used for ablation study in Sec.5.

---

**Example A.4: Prompt for Enriching MATH (ERS)**

You are an answer enricher. I hope you will refine your answers based on the questions and answers provided and avoid jumping your thinking. Pay attention to filling the missing contexts and logical gaps in existing answers. Besides, be careful not to change the original pathways or modify formulas and results, as well as mathematical symbol formats.

**question**: Krista put 1 cent into her new bank on a Sunday morning. On Monday she put 2 cents into her bank. On Tuesday she put 4 cents into her bank, and she continued to double the amount of money she put into her bank each day for two weeks. On what day of the week did the total amount of money in her bank first exceed \$2?

**answer**: The formula for a geometric series is $\frac{a-ar^n}{1-r}$. Taking $a$ to be the initial 1-cent deposit and $n$ to be the number of days Krista had money in her bank so far, we have the inequality

$$\frac{1-2^n}{1-2} \geq 200 \Rightarrow 1 - 2^n \leq -200 \Rightarrow 201 \leq 2^n.$$

The smallest power of 2 that is greater than 201 is $2^8$. Thus, $n = 8$ and $\boxed{\text{Sunday}}$ is 7 days away from day 1.

**enriched answer**: The total amount of money in her bank forms a geometric series. The formula for the sum of a geometric series is $\frac{a-ar^n}{1-r}$, where $a$ is the first term, $r$ is the common ratio, and $n$ is the number of terms. In this case, $a$ is the initial 1-cent deposit, $r$ is 2 (since she doubles the amount each day), and $n$ is the number of days Krista had money in her bank so far. We want to find the smallest $n$ such that the total amount of money in her bank exceeds \$2, or 200 cents. This gives us the inequality:

$$\frac{1-2^n}{1-2} \geq 200 \Rightarrow 1 - 2^n \leq -200 \Rightarrow 201 \leq 2^n.$$

The smallest power of 2 that is greater than 201 is $2^8$. Thus, $n = 8$. Since she started on a Sunday, and there are 7 days in a week, the day of the week when the total amount of money in her bank first exceeded \$2 is $\boxed{\text{Sunday}}$, which is 7 days away from day 1.

**question**: {Q}
**answer**: {A}
**enriched answer**:

