# OpenReview forum: "System-2 Mathematical Reasoning via Enriched Instruction Tuning"
_TMLR — Accepted by TMLR_

### Review · Reviewer_Uuqu · 2025-03-15

**Summary Of Contributions:**

This paper introduces Enriched Instruction Tuning (EIT), which prompts GPT-4 using Enriching with Reasoning Plan (ERP) and Enriching with Reasoning Step (ERS) to generate a dataset that augments human annotations with system 2 reasoning steps. The new augmented dataset is then used to fine-tune a large language model, which shows improvement in performance over other fine-tuning baselines on GSM8K and MATH.

**Audience:**

Yes

**Claims And Evidence:**

Yes

**Requested Changes:**

Summary of weaknesses to address:
- Provide better analysis for the claims that the proposed method mitigates the limitations of LHF and LAIF.
- Provide empirical analysis to Auto-CoT
- Clarify reported results.

**Strengths And Weaknesses:**

**Strengths:**
- The paper’s methodology is motivated by system 2 reasoning.
- The experiments show improvement in performance over baselines.
- The paper is easy to read with good case studies.

**Weaknesses:**

The paper claims to “combine the strengths of LHF and LAIF in a novel way that avoids their limitations”. However, I am not convinced that this is true.
- The limitation of LHF is the prohibitively expensive and impracticality in collecting a high quality human annotated dataset. EIT’s ERS step augments human-annotated instruction-response pairs, which suggests that the method still relies on high-quality human annotation for datasets, thus suffers from the same downsides outlined by the authors.
- The limitation of LAIF is LLM hallucinations. Which the paper claims to mitigate because the “proposed method strictly follows the original response without additional mathematical or symbolic calculations”. However, from my understanding, this constraint is enforced only through prompting, which does not guarantee the elimination of hallucinations. Especially since ERP and ERS generates "implicit steps omitted by humans", which has no clear verification that these steps are factual and not hallucinated.
- Another limitation of LAIF is that they “cannot master system-2 reasoning capabilities”. However, EIT’s ERP step is prompting GPT4 to generate low-level objectives through CoT with few-shot examples, which requires system-2 reasoning.

The paper states that Auto-CoT is the most closely related work, but no experiments are conducted to make empirical comparisons. Instead, Auto-CoT is dismissed for potentially causing hallucinations, without substantive evaluation.

The proposed method does not perform as well as tool-augmented LLMs and the paper does not justify enough why fine-tuning is more favorable than tool-augmented LLMs.

The reported results can be made clearer.
- In Section 3.1, the paper states, “We compared the final answers of the generated data with the original answers to calculate accuracy. Our results show an accuracy rate of 99.7% for the generated answers, whereas the accuracy of GPT-4’s final answers without our EIT prompts is only 71.5%.” However, the meaning and implications of these accuracy figures are unclear.
- The numbers reported in Table 3 are lower than those reported elsewhere in the paper, creating ambiguity about which model achieved 84.1% accuracy.

---

> ### Author Response · Authors · 2025-03-31
> **Response to Reviewer Uuqu (part1)**
>
> ### **Q1** The paper claims to “combine the strengths of LHF and LAIF in a novel way that avoids their limitations”. However, I am not convinced that this is true. ...the method still relies on high-quality human annotation for datasets, thus suffers from the same downsides outlined by the authors...from my understanding, this constraint is enforced only through prompting, which does not guarantee the elimination of hallucinations...EIT’s ERP step is prompting GPT4 to generate low-level objectives through CoT with few-shot examples, which requires system-2 reasoning.
>
> **A:** Thank you for your insightful comments. We appreciate the opportunity to clarify our work and address your concerns.
>
> 1. Annotating detailed and comprehensive data like EIT dataset from scrach is prohibitively expensive and time-consuming. ERS  helps to aleviate this limitation, which only necessitates annotators to provide basic, concise solution steps. These basic steps, while still high-quality, demand significantly less expertise and time from annotators. ERS then enriches these basic annotations to get a smoother reasoning trajectory, thereby lowering the overall cost and effort associated with high-quality human annotation. Consequently, our approach effectively mitigates the primary drawback of LHF.
> 2. Our paper does not claim to guarantee the complete elimination of hallucinations in LAIF. Instead, we aim to mitigate them by adhering closely to human-annotated pathways. By enriching sparse human-annotated steps rather than generating detailed solutions from scratch, our method reduces the complexity of LAIF and mitigates hallucinations.
> 3. EIT significantly lowers the requirement of system-2 reasoning for GPT-4. Compared to obtaining direct responses from GPT-4 or less powerful LLMs, our approach more effectively mitigates hallucinations and improve their capabilities. As shown in a quantitative analysis, we prompted GPT-4 directly to generate responses on the MATH and GSM8K training sets and fine-tuned LLaMA-2 to compare with our EIT method. EIT improves testing accuracy on the MATH dataset from 20.7% to 32.5%, and on GSM8K from 77.5% to 84.1%. This indicates a significant reduction in hallucinations compared to the baseline.
>
> **Table 1: Comparison of testing accuracy (%).**
> | Model | Size | MATH | GSM8K |
> |-------|------|------|-------|
> | LLaMA-2 （responses from GPT-4 without human annotations) | 70B | 20.7 | 77.5 |
> | LLaMA-2 (using our EIT) | 70B | 32.5 | 84.1 |
>
> Consequently, while EIT still requires partial reasoning capabilities for the annotation LLM (i.e. GPT-4 in our experiments), our EIT method enhances these capabilities based on strategic prompting and precollected human annotations. As evidenced in Table 1, our EIT model is superior to those obtained by directly using GPT-4 data, which validate the point that our EIT enhance GPT-4's system-2 reasoning abilities to generate more high-quality data for finetuning.
>
> ### **Q2** The paper states that Auto-CoT is the most closely related work, but no experiments are conducted to make empirical comparisons. Instead, Auto-CoT is dismissed for potentially causing hallucinations, without substantive evaluation.
>
> **A:** Thank you for the thoughtful feedback. We have added results on the GSM8K dataset in Table 2 to empirically compare EIT with Auto-CoT, incorporating additional reasoning steps. Our findings indicate that EIT surpasses Auto-CoT by 5.3%.
>
> **Table 2: Comparison of testing accuracy (%).**
> | Model | GSM8K |
> |-------|-------|
> | Auto-CoT(Add Reasoning Step) [1] | 78.8 |
> | EIT | 84.1 |
> [1] Jin, Mingyu, et al. "The impact of reasoning step length on large language models." arXiv preprint arXiv:2401.04925 (2024).
>
>
> ### **Q3** The proposed method does not perform as well as tool-augmented LLMs and the paper does not justify enough why fine-tuning is more favorable than tool-augmented LLMs.
>
> **A:** Thank you for your feedback. We would like to clarify that our paper does not emphasize that EIT is more favorable than tool-augmented LLMs. Instead, we believe that both approaches have their own strengths and are suitable for different scenarios.
>
> Both approaches exhibit complementary advantages. Fine-tuning is well-suited for semantic analysis, planning, and abstract reasoning (e.g., commonsense reasoning), but they struggle with precise computation, symbolic manipulation, and algorithmic processing. Conversely, tool-augmented LLMs excel in rigorous operations and can outsource intricate calculations to specialized tools like equation solvers. [2]
>
> Therefore, the choice between fine-tuning and tool-augmentation should be based on the specific requirements of the task at hand, rather than a preference for one method over the other.
>
> [2] Gou, Zhibin, et al. "Tora: A tool-integrated reasoning agent for mathematical problem solving." arXiv preprint arXiv:2309.17452 (2023).

---

> > ### Author Response · Authors · 2025-03-31
> > **Response to Reviewer Uuqu (part2)**
> >
> > ### **Q4** The reported results can be made clearer. In Section 3.1...However, the meaning and implications of these accuracy figures are unclear. The numbers reported in Table 3 are lower than those reported elsewhere in the paper, creating ambiguity about which model achieved 84.1% accuracy.
> >
> > **A:** Sorry for any confusions.
> > 1. We compared the final answers of the generated responses with the original answers to calculate accuracy. Our results show an accuracy rate of 99.7% for the generated answers, whereas the accuracy of GPT-4's direct responses without our EIT prompts is only 71.5%. These findings indicate that our EIT approach substantially enhances the quality of the generated responses and mitigates issues related to hallucinations which lead to wrong final answers.
> >
> > 2. As we illustrated in Section 4.2, to make a fair comparison, we only use 15k as the training set for ablation study to compare with the baseline, which use 15k(7.5k from MATH and 7.5k from GSM8K) for training. To make a fair comparison with other SOTA methods using larger datasets (e.g., MetaMath for 395k, MAmmoTH for 260k), we expand EITMath to 70k and get a better model with the accuracy of 84.1% on GSM8K dataset.

---

> > > ### Comment · Reviewer_Uuqu · 2025-04-12
> > >
> > > Thank you for the response. I recommend that the authors clarify the points I raised in the next version of the paper.

---

> > > > ### Author Response · Authors · 2025-04-12
> > > > **Response to Reviewer Uuqu**
> > > >
> > > > Thank you for your valuable feedback and for acknowledging our response. We will definitely incorporate the points you raised into the next version of our paper.

---

### Review · Reviewer_kvPW · 2025-03-16

**Summary Of Contributions:**

This paper introduces Enriched Instruction Tuning (EIT) for complex mathematical reasoning inspired by system-2. The key idea of EIT is leveraging the synergy between human and AI feedback to produce enriched instruction datasets consisting of fine-grained reasoning trajectories. Concretely, EIT comprises two main steps: (1) generation of enriched instruction dataset and (2) LLM fine-tuning, which we will introduce in the following sections respectively. The enriched instruction datasets are utilized to fine-tune open-source LLMs (i.e., LLaMA-2), thereby boosting their inherent ability to execute system-2 mathematical reasoning without any usage of external tools. Experimental results on MATH and GSM8K show that LLMs fine-tuned by EIT achieve competitive performance compared with fine-tune, prompting-based, and tool-augmented baselines.

**Audience:**

Yes

**Claims And Evidence:**

Yes

**Requested Changes:**

More latest LLMs should be involved to justify if the improvements hold on top of more powerful LLMs as well as o1/R1-style reasoning LLMs.

**Strengths And Weaknesses:**

Strengths:

1. The paper is well-written. The motivation is generally clear and the proposed approach is easy to understand and reproduce.

2. It is reasonable to enrich the sparse human-annotated dataset with high-level plans as well as fine-grained execution steps. The implementation recipe is thoughtful --- the proposed method strictly follows the original response without involving additional mathematical or symbolic calculations, to maintain the accuracy of human annotations while significantly mitigating LLMs’ hallucinations on long-horizon reasoning trajectories.

3. There are some interesting observations which may be beneficial for related studies: (i) the perplexity is inversely proportional to accuracy; (ii) more fine-grained reasoning trajectories consistently lead to higher testing accuracy.

Weaknesses:

1. My major concern is that only LLaMA-2 is used as the base model for fine-tuning. As the enriched datasets are feasible to be adopted for a wide range of LLMs. It is not clear whether the improvements hold on more powerful LLMs such as Llama-3 as well as the latest o1/R1-style augmented reasoning models. I like the idea of enriching the dataset with ERP and ERS. It would be inspiring to identify the effectiveness of the approach on more latest LLMs.

2. This paper has mentioned system-2 for several times and implied that existing studies lack system-2 deliberate reasoning abilities. On the one hand, the claim might be misleading as recent reasoning models have been widely adopted long CoT to induce fine-grained reasoning steps. On the other hand, though this paper claims to leverage the synergy of human and AI feedback, the approach is more likely to be a data augmentation approach with abstract planning and more fine-grained reasoning steps based on top of the original human annotations, instead of the interactive improvements between human and AI collaboration.

---

> ### Author Response · Authors · 2025-03-31
> **Response to Reviewer kvPW**
>
> ### **Q1**  1.My major concern is that only LLaMA-2 is used as the base model for fine-tuning...I like the idea of enriching the dataset with ERP and ERS. It would be inspiring to identify the effectiveness of the approach on more latest LLMs.
>
> **A:** Thank you for your insightful suggestions. Due to the limited rebuttal period and computing resources, we add two latest LLMs, including Llama-3 and DeepSeek-Math. As shown in the following table, Llama-3-8B achieves above 8% improvement on two benchmarks with our EIT method. Remarkably, for a more powerful LLM, DeepSeek-Math-base-7B, our EIT can achieve a 9.4% improvement on GSM8K, further validating that our EIT is beneficial for a wide range of LLMs.
>
> **Table 1: Comparison of testing accuracy (%) on more latest LLMs.**
>
> | Model | MATH | GSM8K |
> |-------|------|-------|
> | Llama-3-8B [1] | 20.3 | 57.2 |
> | EIT(Llama-3-8B) | 28.1 (+8.2) | 65.8 (+8.6) |
> | DeepSeek-Math-base-7B [2] | 36.2 | 64.2 |
> | EIT(DeepSeek-Math-base-7B) | 41.3 (+5.1) | 73.6 (+9.4) |
>
> [1] Grattafiori, Aaron, et al. "The llama 3 herd of models." arXiv preprint arXiv:2407.21783 (2024).
>
> [2] Shao, Zhihong, et al. "Deepseekmath: Pushing the limits of mathematical reasoning in open language models." arXiv preprint arXiv:2402.03300 (2024).
>
>
> ### **Q2**  This paper has mentioned system-2 for several times and implied that existing studies lack system-2 deliberate reasoning abilities. On the one hand, the claim might be misleading ... On the other hand..the approach is more likely to be a data augmentation...
>
> **A:**  Thank you for the thoughtful feedback. We appreciate your insights and would like to clarify our contributions and address the concerns raised:
>
> 1. **Regarding System-2 Reasoning:**
> First of all, we would like emphasize that EIT was developed before the recent large reasoning model such as o1 and r1. While recent advances in long CoT reasoning are impressive, they inherently validate the foresight of our work, highlighting the necessity of hierarchical planning and system-2 thinking in complex reasoning tasks. Crucially, our approach provides a complementary pathway to existing long CoT methods: rather than relying solely on reinforcement learning or distillation from reasoning models, we demonstrate how short human (CoT) annotations can be systematically restructured into long CoT reasoning through our method for reasoning LLM training.
>
> 2. We agree that the statement of data augmentation approach might be more easy to understand for readers and will definitely revise the expressions to avoid potential misunderstandings.

---

### Review · Reviewer_Tp8P · 2025-03-17

**Summary Of Contributions:**

This paper introduces a prompting method to rewrite human-written math data to improve LLM's math ability. More specifically, the idea is to use one prompt (ERP) to generate planning and the other prompt (ERS) to add more details to human-written answers. The method works well empirically and outperforms several strong augmentation or CoT methods on the GSM8k and the MATH benchmark. This work also includes interesting analyses studying different factors (e.g., instruction length, perplexity, etc).

**Audience:**

Yes

**Broader Impact Concerns:**

No ethical concerns.

**Claims And Evidence:**

Yes

**Requested Changes:**

1. Add a detailed analysis (beyond the three case studies shown in the current paper) on the answer quality.
2. Fix issues mentioned in point 2 above.

**Strengths And Weaknesses:**

Strengths:
1. The proposed method is simple and effective. The prompts in this work represent a nice direction to make human responses more learnable while avoiding too much additional noise from model generations.
2. Overall, this is a well-written paper with interesting ablations.

Weaknesses:
1. There is no quantitative analysis on how good these prompts are. I understand that the goal is to avoid adding any "additional mathematical or symbolic calculations" (as mentioned in Sec. 3.1), but the prompts will add details to fill in the gaps in human answers, so some of the prompts will still include errors. It will be useful to do (even at a relatively small scale) a human study to check the quality of rewritten responses. It would also be very valuable if there is any pattern in human responses that the proposed approach is good (or bad) at.
2. There are a few small inclarities about the experiment details:
    1. How do you control the length or the granularity of the outputs for the analysis in Sec. 6.2?
    2. Is the perplexity average or max perplexity over the tokens?
    3. The scaling improvement in Figure A.1 looks quite small. Do you run multiple seeds or compute statistical significance of the improvement?

---

> ### Author Response · Authors · 2025-03-31
> **Response to Reviewer Tp8P**
>
> ### **Q1**  There is no quantitative analysis on how good these prompts are.
>
> **A:** Thank you for your insightful feedback.
>
> 1. **Human Study:** We acknowledge the importance of evaluating the quality of the prompts generated by our method. Due to the large scale of the dataset and limited rebuttal period, we conducted a human study by randomly sampling 100 examples from different types and difficulty levels for manual analysis. Our careful examination revealed that the quality is generally high, with only 2 samples exhibiting minor noise issues. Below is one of  the identified noise cases:
>
> | Problem | Human-Annotated Solution | EIT Solution |
> |--------------|--------------|--------------|
> | What is the units digit of $\\frac{20 \\cdot 21 \\cdot 22 \\cdot 23 \\cdot 24 \\cdot 25}{1000}$? | $\\frac{20\\cdot 21\\cdot 22\\cdot 23\\cdot 24\\cdot 25}{1000} = \\frac{2^2\\cdot 5\\cdot 21\\cdot 2\\cdot 11\\cdot 23\\cdot 2^3\\cdot 3\\cdot 5^2}{2^3\\cdot 5^3} = 2^3\\cdot 3\\cdot 21 \\cdot 11\\cdot 23 \\equiv 2^3\\cdot 3^2 \\pmod{10} \\equiv \\boxed{2}\\pmod{10}$. | ...The units digit of a product depends only on the units digits of the factors, so we can simplify the expression further:$$2^3\\cdot 3\\cdot 21 \\cdot 11\\cdot 23 \\equiv 2^3\\cdot 3 \\pmod{10}.$$...Calculating the units digit of $2^3\\cdot 3^2$ modulo 10:$$2^3\\cdot 3^2 = 8 \\cdot 9 \\equiv 2 \\pmod{10}.$$... |
>
> 2. **Pattern Analysis:**  We performed a pattern analysis on the token length of responses across increasing difficulty levels on the MATH dataset. The results are shown in Table 1.
>
> Our analysis indicates that as the difficulty increases, the length of the responses generated by EIT also increases. Since the responses of simpler questions tend to have more concise explanations and can overlook some steps, the token length for human annotations is relatively short, resulting in larger length ratios for EIT, vice versa. This result demonstrates that EIT can adapt to the difficulty level of the problem and the human annotations, and adaptively enrich necessary details, highlighting the effectiveness of our approach.
>
> **Table 1: Token Length Analysis**
> | Difficulty Level | 1 | 2 | 3 | 4 | 5 |
> |------------------|---|---|---|---|---|
> | Human-Annotated | 57 | 54 | 68 | 161 | 152 |
> | EIT | 240 | 248 | 273 | 345 | 362 |
> | Length Ratio (EIT / Human-Annotated) | 4.2 | 4.6 | 4.0 | 2.1 | 2.4 |
>
>
> ### **Q2**  How do you control the length or the granularity of the outputs for the analysis in Sec. 6.2?...Is the perplexity average or max perplexity over the tokens? ...Do you run multiple seeds or compute statistical significance of the improvement?
>
>
> **A:** Thank you for your thoughtful feedback. We will clarify critical experiment details and improve Section 6.2 accordingly:
>
> 1. **Granularity Control:** We control the granularity of step-by-step reasonings by adjusting the prompts used when requesting responses from GPT-4. This includes specifying a desired number of tokens for the reply and carefully selecting few-shot examples that demonstrate the desired level of detail.
> 2. **Perplexity Metric:** The reported perplexity values represent the average perplexity across all tokens in generated responses.
> 3. **Figure A.1 Analysis:**
> We did not run multiple random seeds for Figure 1 due to the expensive experiment cost. To address this, we conducted additional experiments using five different random seeds. We evaluated the training metrics across four different data sizes and calculated the mean and standard deviation of the accuracy, as shown in Table 2. Despite these fluctuations, the model's accuracy showed a consistent improvement across different data sizes.
>
> **Table 2: Accuracy Across Different Data Sizes with Multiple Random Seeds**
> | Data Size | 20k | 40k | 80k | 120k |
> |-----------|-----|-----|-----|------|
> | Accuracy-Mean (%) | 30.20 | 30.66 | 31.46 | 31.98 |
> | Accuracy-Std (%) | 0.33 | 0.28 | 0.41 | 0.30 |

---

### Decision · Action_Editor_Up6r · 2025-08-13

**Recommendation:** Accept with minor revision

**Audience:**

Yes

**Audience Explanation:**

Math problem solving with LLM is one of frontiers for LLM reasoning research

**Claims And Evidence:**

Yes

**Claims Explanation:**

This paper introduces prompt engineering method to rewrite human-written math data to improve LLM's math ability. The key idea is to use one prompt (ERP) to generate planning and use the other prompt (ERS) to add more details to human-written answers. The method works well empirically and outperforms several strong augmentation or CoT methods on the GSM8k and the MATH benchmark. This work also includes interesting analyses studying different factors (e.g., instruction length, perplexity, etc).

After the rebuttal, the paper received three "Learning Accept". The AE suggested the authors to prepare a final version to make Sec. 3.1 clearer following Reviewer Uuqu's suggestions.